# Anabolic–Androgenic Steroid Abuse among Gym Users, Eastern Province, Saudi Arabia

**DOI:** 10.3390/medicina57070703

**Published:** 2021-07-10

**Authors:** Walied Albaker, Ali Alkhars, Yasir Elamin, Noor Jatoi, Dhuha Boumarah, Mohammed Al-Hariri

**Affiliations:** 1Department of Internal Medicine, College of Medicine, Imam Abdulrahman Bin Faisal University, P.O. Box 2114, Dammam 31451, Saudi Arabia; wialbakr@iau.edu.sa (W.A.); aakhars@iau.edu.sa (A.A.); yaelamin@iau.edu.sa (Y.E.); najatoi@iau.edu.sa (N.J.); dohanahar@gmail.com (D.B.); 2Department of Physiology, College of Medicine, Imam Abdulrahman Bin Faisal University, P.O. Box 2114, Dammam 31451, Saudi Arabia

**Keywords:** anabolic steroids, gym members, male, gym, abuse

## Abstract

*Background and Objectives*: The main aim of the present study was to assess the use of androgenic–anabolic steroids (AAS) and to investigate its potentially unfavorable effects among gym members attending gym fitness facilities in Eastern Province, Saudi Arabia. *Materials and Methods*: A cross-sectional questionnaire-based study was carried out during the summer of 2017. Male gym users in the Eastern Province region of Saudi Arabia were the respondents. Information on socio-demographics, use of AAS, knowledge, and awareness about its side effects were collected using a self-administered questionnaire. *Results:* The prevalence of AAS consumption among trainees in Eastern Province was 21.3%. The percentage was highest among those 26–30 years of age (31.9%), followed by the 21–25 (27.4%) (*p* = 0.003) age group. Participants in the study were not aware of the potential adverse effects of AAS use. Adverse effects experienced by 77% of AAS users include psychiatric problems (47%), acne (32.7%), hair loss (14.2%), and sexual dysfunction (10.7%). Moreover, it appears that trainers and friends are major sources (75.20%) for obtaining AAS. *Conclusion:* AAS abuse is a real problem among gym members, along with a lack of knowledge regarding its adverse effects. Health education and awareness programs are needed not only for trainees, but also for trainers and gym owners as they are reportedly some of the primary sources of AAS.

## 1. Introduction

Androgenic–anabolic steroids (AAS) are synthetically occurring products of the male sex hormone (Testosterone) [1]. They have two major effects which are anabolic and androgenic in nature [2]. The anabolic effect leads to decreased body fat and increases bone density and skeletal muscle mass, as well as stimulating erythropoiesis [3]. The androgenic effects are associated with the development of male sexual characteristics [4]. They have a significant effect on athletic performance [1].

A higher level of AAS in the body can lead to several psychological and physical complications. Examples of the physical problems that AAS can lead to are high blood pressure, atherosclerosis, myocardial infarction, cardiac hypertrophy, fluid retention, jaundice, acne, and hepatic tumors [1,5]. Psychiatric problems such as aggressiveness, euphoria, irritability, and mood disturbance can occur. Furthermore, AAS can indeed cause reduced sperm count, shrinking of the testicles, infertility, baldness, and the development of prostate and breast cancers [6].

Most countries permit the use of AAS to treat medical conditions by prescription [7]. Studies show an increasing rate of AAS use among athletes worldwide. However, the World Anti-Doping Agency lists AAS as prohibited substances [8].

Many bodybuilders turn to these medications to increase muscle size, strength, and overall efficiency with less effort over a shorter time rather than relying on physical exercise and a healthy diet alone [9].

Data show that AAS abuse is more prevalent in Brazil, Western countries, and the Middle East, and is less prevalent in Asia and Africa [10]. A few studies have evaluated the prevalence and awareness about the use of AAS among bodybuilders in Saudi Arabia. Previously reported data found that AAS ranked the highest among Saudi athletes who tested positive for prohibited agents [5]. Another study evaluating AAS abuse among bodybuilders in the southern province of Saudi Arabia (Jazan) revealed a lifetime prevalence of 31% [11]. 

The prevalence of AAS abuse was lower in the 316 male gym users in the northwestern region of Saudi Arabia (11.4%) [12]. Meanwhile, many studies have concluded that a lack of knowledge is the most likely cause regarding the use of AAS and its adverse effects on Saudi gym members. These studies also recommended a national awareness program in the central (Riyadh) [5], western (Jeddah) [12], and southern (Jazan) [11] regions of Saudi Arabia. Therefore, the present study sought to assess the prevalence of androgenic–anabolic steroid (AAS) use, and to investigate motivations for use and knowledge of its potentially unfavorable effects among gym center visitors in Eastern Province, Saudi Arabia.

## 2. Materials and Methods

A cross-sectional survey-based study was conducted on participants visiting gymnasiums in the Eastern Province (Al-Hufof, Ad-Dammam, Al-Khobar, Al-Qatif, Al-Jubail, and Saihat), Kingdom of Saudi Arabia, during the summer of 2017.

The calculation of the sample size was based on a prevalence of AAS of 50%, with a 95% confidence interval (CI). The inclusion criteria were the following: male, older than 18 years, and willingness to take part. We excluded repeated registration in multiple centers or incomplete responses.

The Ethics Committee of Imam Abdulrahman Bin Faisal University approved the research proposal and questionnaire (IRB-2018-01-174, approval date, 27 September 2018). All the participants agreed to participate and signed a consent form before entering into the study.

### 2.1. Questionnaire

A self-administered questionnaire was used for data collection. The questionnaire was designed based on previous similar studies reported in the literature on the same topic [12,13]. A pilot study was then carried out to determine the reliability and validity of the questionnaire. The feedback was analyzed and a finalized questionnaire was created accordingly. The final version of the questionnaire consisted of questions on (1) sociodemographic characteristics; (2) overall knowledge about the most commonly used and adverse effects of anabolic steroids; (3) prevalence of side effects experienced by users; and (4) the practices and patterns of AAS use. A single set of questionnaires was distributed (Arabic version).

### 2.2. Statistical Analysis

The statistical analysis was performed using SPSS software version 23.0 (SPSS Inc., Chicago, IL, USA). Descriptive statistics were used to explain the categorical and outcome items. A comparison between subgroups was made using Pearson’s Chi-square test. A *p*-value less than 0.05 was used to indicate statistical significance.

## 3. Results

### 3.1. Sociodemographic Characteristics

Nearly 597 eligible gym members were invited to participate in the present study. A total of 541 participants answered the questionnaire and were included in the study according to the calculated sample size, with an overall response rate of 90.6%. Table 1 shows the characteristics of the study sample. Most of the users (67.1%) belong to the 21–25 age group. However, the present data showed that 72.3% of the studied gym members were currently employed and received higher education, i.e., bachelor’s degree or higher (67.1%). Moreover, the majority of the study participants were single (61%), and half of them (50.8%) drew income less than 5000 Saudi Riyal (SR) (Table 1).

### 3.2. Practices Associated with Androgenic–Anabolic Steroid Consumption

As shown in Figure 1, the most commonly used AAS in our study across all ages were Anavar (61.9%), Dianabol (46%), and Deca Durabolin (45.1%).

Nearly 64.6% of gym members had been consuming AAS for more than 5 years, while 20.4% had been using steroids for three years or more (Figure 2).

The majority of the AAS users (77%) reported side effects, and 47% experienced psychiatric problems, including depression, insomnia, and lower appetite. Acne was reported in 32.7% and hair loss in 14.2% of participants, as shown in Table 2.

The prevalence of AAS users in this study was 21.3%. Furthermore, it was higher in the age group of 26–30 (31.9%), followed by the 21–25 group (27.4%). The majority of them were employed (82.3%) with low (less than 5000 SR) income (41.6%), as presented in Table 3. According to Figure 3, it appears that trainers and friends are a major source for obtaining AAS.

Approximately 63.7% of participating gym members observed an increase in activity, while 76.1% reported an increase in power after using AAS. However, 88.5% of users reported fast muscle mass gain (Figure 4).

### 3.3. Knowledge about and Attitude towards Androgenic–Anabolic Steroid Consumption

Surprisingly, the participants in our study were not aware of the potential adverse effects of AAS use. However, the majority of the AAS users had adequate knowledge of the adverse effects of AAS compared with nonusers. Approximately 38.1% of participants were aware that using AAS may lead to increased blood pressure, hair loss (34.5%), and acne (33.6%). Meanwhile, the nonusers expressed more knowledge of infertility (48.69%). About 18.6% of AAS users think that using AAS without a medical prescription is legal, while 15% did not know if it was legal to use ASS with a medical prescription or not. Among users, 66.4% believe it is illegal and still use it (Table 4).

## 4. Discussion

In this study, we evaluated the prevalence, practices, knowledge, and attitudes of gym members in Eastern Province, Saudi Arabia, using AAS. The prevalence of AAS users among males attending gym centers was as high as 21.3%. These findings are in accordance with other studies conducted in the region (Kuwait and the UAE) [14,15]. Furthermore, two local studies were done in 2016 and 2017, in Riyadh [13] and Jazan [12], respectively. Both studies show an approximately 10% higher prevalence rate.

There are many reasons behind the prevalence of AAS consumption among gym members in the Eastern Province, such as competition among athletes to quickly build muscle mass. Respondents believed that the muscle growth advantages of AAS outweighed its adverse harmful effects [15], as the use of AAS makes lifting heavy objects easier [5]. Even some trainees recommended allowing the use of AAS for enhancing performance [16].

A local study found that the majority (77%) of athletes who self-declared AAS use and were aware of their adverse effects would still recommend them to friends [13]. Another reason for the high use rates could be due to the availability of AAS in gym centers and from trainers and friends. As reported recently, the prevalence of AAS consumption was highly influenced by its availability [16].

The highest prevalence of AAS usage in our study was among the 21–30 age group. This result is different from what was reported in Kuwait and the Middle East and North Africa (MENA) regional studies [15]. Similar profiles have been reported in other local studies, with the majority being the age group of 25 to 29 years [5,11].

In spite of the significant level of knowledge expressed in the study sample, the findings indicate a wide range of practices within the studied gym members. This suggested inadequate insight and a lack of healthcare professionals specializing in sports science who may be able to popularize an understanding of the wide range of adverse effects related to AAS use [7].

Our study revealed a negative association between monthly income and level of education with the prevalence of AAS use, which means that knowledge and education were independent of AAS use. Remarkably, these findings were also observed in previous studies that were performed in the region [5].

The most common types of AAS used in our findings were Anavar, Dianabol, and Deca Durabolin. This differed from previously reported data in another region of Saudi Arabi [17]. This could reflect the availability and preference for AAS in Saudi Arabia.

## 5. Conclusions

Based on this study, the prevalence of AAS users in Eastern Province, Saudi Arabia is high, which reflects the fact that gym members are at a higher risk of using AAS. Therefore, Regional Health Authorities in the Eastern Province region, Saudi Arabia, should urgently take measures to alleviate the potential adverse implications of AAS consumptions among young adults by using social media as an educational tool, or by distributing informative leaflets among AAS users.

## 6. Limitation

Although this is the first study conducted on gym members in the Eastern Province regarding AAS consumption, it had some limitations. Notably, it was only performed among male gym members, no blood work was included to investigate potential adverse implications of AAS, and finally, we used a self-reported questionnaire.

## Figures and Tables

**Figure 1 medicina-57-00703-f001:**
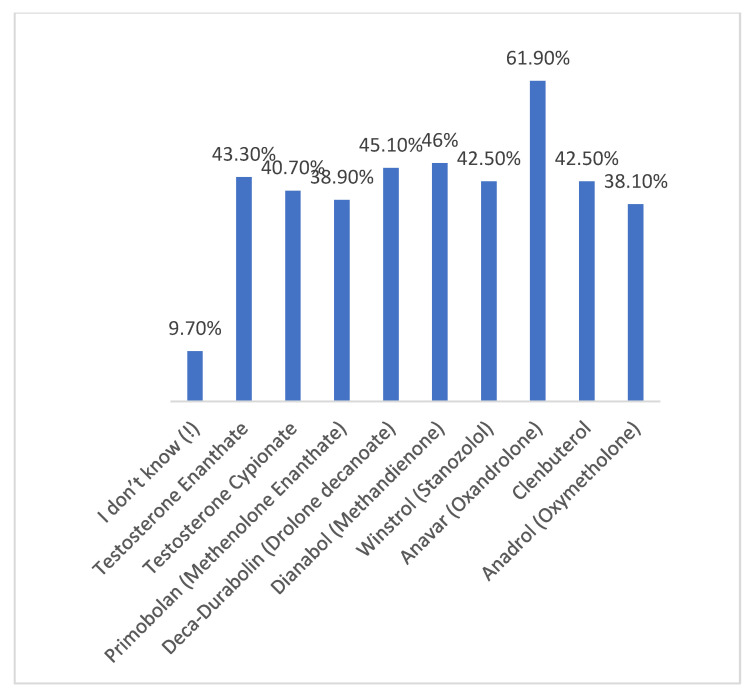
The prevalence of the most commonly used Androgenic–Anabolic Steroids (AAS) brands (generic).

**Figure 2 medicina-57-00703-f002:**
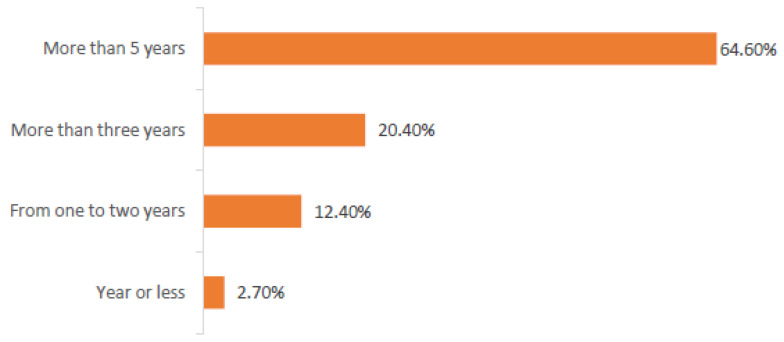
The relationship Between Using of Steroids and Years of Practice.

**Figure 3 medicina-57-00703-f003:**
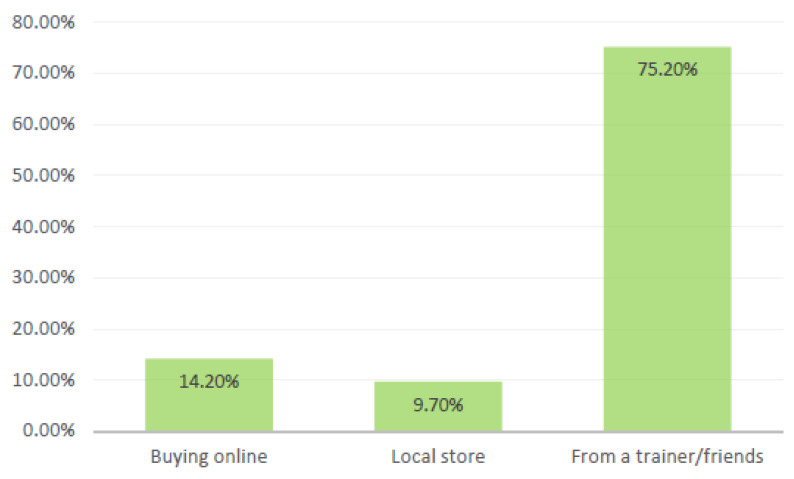
Sources of steroids.

**Figure 4 medicina-57-00703-f004:**
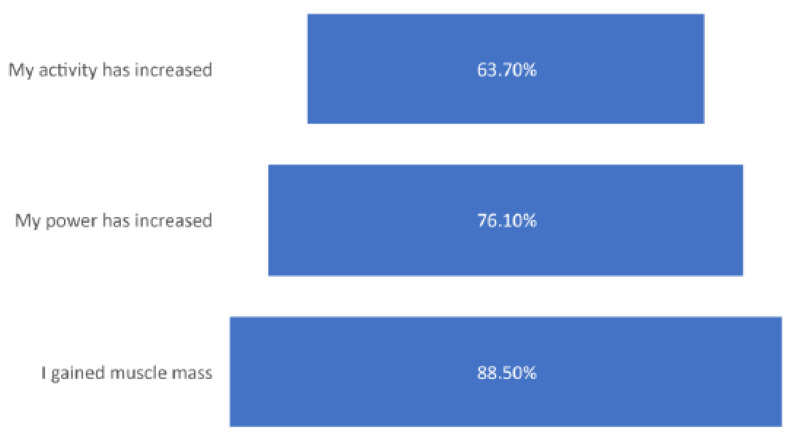
Reported advantages of using anabolic steroids.

**Table 1 medicina-57-00703-t001:** Sociodemographic characteristics (*n* = 541).

Characteristic	*N*	%
Age Group		
Less than 18	11	2
18–20	53	9.8
21–25	184	67.1
26–30	161	29.8
31–35	66	12.2
35 and older	66	12.2
Occupation		
Employed	391	72.3
Unemployed	150	27.7
Level of Education		
Intermediate or Lower	23	4.3
High School	155	28.7
Higher Education	363	67.1
Social Status		
Single	330	61
Married	211	39
Monthly Income Saudi Riyal (SR)		
Less than 5000	275	50.8
5000–10,000	173	32
More than 10,000	83	15.3

**Table 2 medicina-57-00703-t002:** The prevalence of side effects experienced by users.

Side Effect	%
Any Side effect	77%
Acne	32.7%
Hair loss	14.2%
Breast Enlargement	9.7%
Psychiatric problems	47%
Sexual Dysfunction	10.6%
Chest pain	2.7%

**Table 3 medicina-57-00703-t003:** The prevalence of anabolic steroid use according to selected characteristics.

Characteristic	Users/Total	Prevalence%
Prevalence	113/541	21.3
Age Group		
Less than 18	0/11	0
18–20	5/53	4.4
21–25	31/184	27.4
26–30	36/161	31.9
31–35	19/66	16.8
35 and older	22/66	19.5
Occupation		
Employed	93/391	82.3
Unemployed	20/150	17.7
Level of Education		
Intermidate or lower	3/23	2.7
High School	35/155	31
Higher Education	75/363	66.4
Social Status		
Single	66/330	58.4
Married	47/211	41.6
Monthly Income		
Less than 5000	47/275	41.6
5000–10,000	43/173	38.1
More than 10,000	23/83	20.4

**Table 4 medicina-57-00703-t004:** Knowledge about the adverse effects of anabolic steroids.

Parameter	Using Steroids			
	Users		Non-Users	
Acne	38	33.6%	86	20.1%
Heart Problems	19	16.8%	100	23.4%
Hair Loss	39	34.5%	86	20.1%
Increase Blood Pressure	43	38.1%	106	24.8%
Infertility	27	23.9%	208	48.6%
Liver Problems	29	25.7%	113	31.1%
There is No harm	7	6.2	0	0%
I don’t know	24	21.2	160	37.4%
Legal	21	18.6%	29	6.8%
Illegal	75	66.4%	294	68.7%
I don’t know	17	15%	105	24.5%

## Data Availability

The data used to support the findings of this study are available from the corresponding authors upon reasonable request.

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
