# Peer review of "Anabolic–Androgenic Steroid Abuse among Gym Users, Eastern Province, Saudi Arabia"

_medicina, 2021, doi:10.3390/medicina57070703_

Round 1
Reviewer 1 Report
The authors conducted a questionnaire-based survey of gym users within the Eastern Province of Saudi Arabia. Overall, the major issue is that the Materials and Methods are not very well described. This makes it difficult to even evaluate whether the Results and Conclusions are supported by the data. Beyond that, the authors make conclusions and comments that that tend to be overinterpretations of the data provided.
Some examples of further deficiencies in the study are provided below, but this reviewer cautions that these are just examples. There are just too many to list:
1, The authors seem to use the term body builder, athlete and visitor to a gym interchangeably. For clarity, define the term used—what is a ‘body-builder’ and who defined that? The study participants? Those administering the questionnaire? My sense is that the authors just surveying people attending gyms. If that’s the case, describe them as gym members and eliminate suggestive terms such as body-builder or athlete unless the subject actually were surveyed to self-describe themselves as such. If they were surveyed, then present the data.
2, The questionnaire itself, translated or original, is not provided to the reviewer and it is not at all clear if all the questions are presented in the Tables.
3, Pertinent details for evaluating the pertinence of any questionnaire are missing, including what % of those offered the questionnaire declined to participate?
4, Income (Table 1) is difficult to assess for those unfamiliar with Saudi culture. Please describe in relationship to the median income within age group within the geographic study area.
5, Figure 1 shows the types of AASs used within the study group. Presumably, this is a % only within those who reported abuse. With no explanation in the text and minimal materials and methods, this is not as clear as it should be for a scientific publication. Is that AAS use prevalence over the lifetime of the AAS user or limited to within a certain time period (for example, within the past year). This is just one example of many details absent throughout the manuscript.
5, There are many other items that do not make much sense in the absence of detail. As just one other example, in Table 3, the authors ascribe P-values to items like prevalence of AAS abuse amongst the study population. This implies that they compared prevalence of abuse against some control population, which is not described anywhere.
6, There are glib statements, such as that in the conclusions section that “These awareness program’s companions must not be exclusive to trainees only, shopping malls, sport clubs, schools, trainers, gym owners and primary health care centers must be covered.’ If there are studies with hard data to support that solution, it is not presented or apparently studies in the current survey. Nor is data cited elsewhere to support that as a discussion point. Indeed, this sounds more like a hypothesis rather than a well-defined conclusion made from the data collected and presented.
Author Response
1, The authors seem to use the term body builder, athlete and visitor to a gym interchangeably. For clarity, define the term used—what is a ‘body-builder’ and who defined that? The study participants? Those administering the questionnaire? My sense is that the authors just surveying people attending gyms. If that’s the case, describe them as gym members and eliminate suggestive terms such as body-builder or athlete unless the subject actually were surveyed to self-describe themselves as such. If they were surveyed, then present the data.
Response
Thank you so much it has been corrected
2, The questionnaire itself, translated or original, is not provided to the reviewer and it is not at all clear if all the questions are presented in the Tables.
Response
Thank you. We have described the components of the questioner at the method section and all components of the used survey are described in the four tables and figures
3, Pertinent details for evaluating the pertinence of any questionnaire are missing, including what % of those offered the questionnaire declined to participate?
Response
I would like to thank the reviewer for this valid point. I have now explained the pertinent details of the study sample.
4, Income (Table 1) is difficult to assess for those unfamiliar with Saudi culture. Please describe in relationship to the median income within age group within the geographic study area.
Response
I am sorry, this point is not clear to me. I don’t understand what is needed exactly.
Do you ask to insert new data on the same table or described this point separately in a different table or to discuss age group income in Saudi with the references? However, the majority of the study participants are with low income, because most of them either students or recent graduates.
5, Figure 1 shows the types of AASs used within the study group. Presumably, this is a % only within those who reported abuse. With no explanation in the text and minimal materials and methods, this is not as clear as it should be for a scientific publication. Is that AAS use prevalence over the lifetime of the AAS user or limited to within a certain time period (for example, within the past year). This is just one example of many details absent throughout the manuscript.
Response
Thank you for this comment. Actually, this survey aimed to study the overall life time (now it has been inserted and edited in the manuscript by the MDPI English Editin).
5, There are many other items that do not make much sense in the absence of detail. As just one other example, in Table 3, the authors ascribe P-values to items like prevalence of AAS abuse amongst the study population. This implies that they compared prevalence of abuse against some control population, which is not described anywhere.
Response
Thank you. p values have been delated as per both reviewers’ requests.
6, There are glib statements, such as that in the conclusions section that “These awareness program’s companions must not be exclusive to trainees only, shopping malls, sport clubs, schools, trainers, gym owners and primary health care centers must be covered.’ If there are studies with hard data to support that solution, it is not presented or apparently studies in the current survey. Nor is data cited elsewhere to support that as a discussion point. Indeed, this sounds more like a hypothesis rather than a well-defined conclusion made from the data collected and presented.
Response
Thank you so much, it has been deleted

Reviewer 2 Report
I want to thank the author for a nice article. I have no comments on the content or design of the studyAuthor Response
I want to thank the author for a nice article. I have no comments on the content or design of the study
Response
Thank you so much for your support

Reviewer 3 Report
Thank you for the opportunity to review this paper. It contains a fairly large sample of gym goers in the Eastern Province of Saudi Arabia.
The manuscript needs to be edited for clarity as well as grammar and spelling (for example, see drug names in Figure 1).
In the Method, I was not sure what the authors meant by the prevalence of AAS being 50%. That seems quite high. Additionally, when the authors begin to report the results, it would be helpful to identity the number of steroid users out of 541 gym goers. The paragraph addressing prevalence of AAS use on page 5 of 9 should be moved up in the Results.
Does there need to be a p-value included for prevalence of use? That is usually a descriptive percentage. Additionally, age group shows a p-value, but chi-square analysis should not be performed when cells have zero observations. Table 4 has the same issues. Differences between users and nonusers do not necessarily need a p-value, as the percentages are the important indicators.
I also would avoid using the terms "gym goers" and "bodybuilders" as synonymous. They typically are two different sets of people with different goals. Also make sure to be specific in identifying percentages, such as whether they reflect the overall sample or just steroid users.
Author Response
Comments and Suggestions for Authors
Thank you for the opportunity to review this paper. It contains a fairly large sample of gym goers in the Eastern Province of Saudi Arabia.
The manuscript needs to be edited for clarity as well as grammar and spelling (for example, see drug names in Figure 1).
Response
Thank you, the manuscript has been, edited by MDPI English Editing, and the figure has been corrected.
In the Method, I was not sure what the authors meant by the prevalence of AAS being 50%. That seems quite high. Additionally, when the authors begin to report the results, it would be helpful to identity the number of steroid users out of 541 gym goers. The paragraph addressing prevalence of AAS use on page 5 of 9 should be moved up in the Results.
Response
Thank you so much for this point. The 50% used for a given level of accuracy (the worst case) to determine a general level of accuracy for a sample.
https://www.surveysystem.com/sscalc.htm
Moreover, I did the needed correction at the method and result sections.
Does there need to be a p-value included for prevalence of use? That is usually a descriptive percentage. Additionally, age group shows a p-value, but chi-square analysis should not be performed when cells have zero observations. Table 4 has the same issues. Differences between users and nonusers do not necessarily need a p-value, as the percentages are the important indicators.
Response
Thank you so much. I have delated the p value as per both reviewers’ requests.
I also would avoid using the terms "gym goers" and "bodybuilders" as synonymous. They typically are two different sets of people with different goals. Also make sure to be specific in identifying percentages, such as whether they reflect the overall sample or just steroid users.
Response
Thank you so much. I have described them as gym members as suggested by the other reviewer as well. Moreover, I have checked for percentage of the overall samples and users

Round 2
Reviewer 3 Report
Thank you for making the suggested edits.